# Detection of pulmonary nodules based on a multiscale feature 3D U-Net convolutional neural network of transfer learning

**Siyuan Tang**, **Min Yang**, **Jinniu Bai**\*

Baotou Medical College, Inner Mongolia University of Science and Technology, Baotou, China

\* baijinniu@163.com

**Data Availability Statement:** All DICOM files are available from the LUNA16 and tianchi database. http://academictorrents.com/collection/luna-lung-nodule-analysis-16—isbi-2016-challenge https://tianchi.aliyun.com/dataset/

## Abstract

A new computer-aided detection scheme is proposed, the 3D U-Net convolutional neural network, based on multiscale features of transfer learning to automatically detect pulmonary nodules from the thoracic region containing background and noise. The test results can be used as reference information for doctors to assist in the detection of early lung cancer. The proposed scheme is composed of three major steps: First, the pulmonary parenchyma area is segmented by various methods. Then, the 3D U-Net convolutional neural network model with a multiscale feature structure is built. The network model structure is subsequently fine-tuned by the transfer learning method based on weight, and the optimal parameters are selected in the network model. Finally, datasets are extracted to train the fine-tuned 3D U-Net network model to detect pulmonary nodules. The five-fold cross-validation method is used to obtain the experimental results for the LUNA16 and TIANCHI17 datasets. The experimental results show that the scheme not only has obvious advantages in the detection of medium and large-sized nodules but also has an accuracy rate of more than 70% for the detection of small-sized nodules. The scheme provides automatic and accurate detection of pulmonary nodules that reduces the overfitting rate and training time and improves the efficiency of the algorithm. It can assist doctors in the diagnosis of lung cancer and can be extended to other medical image detection and recognition fields.

## Introduction

Lung cancer is one of the most common malignant tumors around the world. Early detection can significantly improve the survival rate of patients [1]. The results of the 2011 national lung cancer screening test show that CT detection can significantly reduce the mortality rate of the lung cancer high-risk population by approximately 20%, which confirms the great value of CT in the detection of lung cancer [2]. Pulmonary nodules are the most common form of lung cancer in CT images. Accurate detection of pulmonary nodules is the key to early detection of lung cancer [3].

At present, experts and scholars have proposed many effective methods for detecting pulmonary nodules. The traditional methods first extract the grayscale and texture features of

**Funding:** This work is funded by Inner Mongolia Natural Science Foundation (grant numbers: 2020MS06001). Baotou Medical and health science and Technology Project (grant numbers: Wsjj2019040). Baotou medical college "learning program", "learning program" and "practice program" in 2018 (grant numbers: 2018BYWWJ-ZX-04); Innovation and entrepreneurship training program for college students of Baotou medical college in 2019 (grant numbers: BYDCXL-201922); Bud program of Baotou medical college in 2019 (grant numbers: 2019BYJJ-HL-11).

**Competing interests:** The authors declare that no competing interests exist.

pulmonary nodules; then, the extracted features are classified and trained by an (Support Vector Machine, SVM) and other classifiers to detect pulmonary nodules. Traditional methods require that the characteristics of pulmonary nodules need to be selected artificially; then, algorithms are written to detect pulmonary nodules based on these features [4]. However, the characteristics of artificial selection are not representative. In the process of detection, some nonconforming pulmonary nodules cannot be detected, which will lead to misdiagnosis, prolong the disease and even threaten the patient's life.

Deep learning is an important branch of machine learning and has developed rapidly in recent years. Deep learning can automatically extract multilevel features hidden in data. It has been successfully applied in speech recognition, image recognition, classification and segmentation and other fields [5]. The convolutional neural network is the most successful network model in deep learning. It does not need manual design and can automatically learn based on existing data and can extract image features. It is suitable for the large data volume of medical image processing [6]. The large amount of available data and efficient graphics processor computing power enable the convolutional neural network to obtain good achievements in segmentation and recognition of the brain, cells, lungs, etc. [7].

In the detection method of pulmonary nodules, researchers usually use the deep learning method to extract a large number of features on the 2D surface of pulmonary nodules and obtain detection results. When the image resolution is high and the contrast is obvious, the detection rate of pulmonary nodules attains an ideal result [8]. Images obtained by medical devices encounter interference from various noise information in the outside world. Moreover, the features of pulmonary nodules exist in 3D form, and the characteristic information of pulmonary nodules cannot be completely extracted in 2D space. To fully consider the influence of noise and 3D spatial information of pulmonary nodules, this paper proposes an algorithm for the detection of pulmonary nodules in CT images based on the 3D U-Net convolutional neural network. After 3D sample training, the model can more fully extract original spatial information of nodules and generate more representative features. At the same time, considering the diversity of pulmonary nodules' size and characteristics, the multiscale feature structure in the 3D U-Net convolutional neural network is proposed to detect pulmonary nodules. This scheme adds a dense network to the network structure [9]. The lower layers of pulmonary nodules can be more fully obtained, and expansion convolution is introduced in the first three layers of the network. A multiscale feature detection model is constructed that improves the expressive ability of network features and solves the problem of the low detection rate of irregular and small pulmonary nodules.

The detection of pulmonary nodules by a deep convolutional neural network requires a large amount of data with labels to support the training process. To establish a reliable target dataset in a small number of labeled lung source data and a large number of related lung auxiliary data, In this paper, the transfer learning method is introduced to assist the detection of pulmonary nodules and can solve the difficult problem of manually annotating data [10]. Transfer learning is used to train dataset, and the well-trained deep model is applied to other image datasets for feature extraction. It can improve the system adaptability and detection accuracy and solve the problem of the uneven distribution of the lung dataset. In conclusion, this paper proposes a multiscale 3D U-Net convolutional neural network of transfer learning to detect pulmonary nodules.

## Related work

A convolutional neural network is a deep network structure based on the deep learning method; it has two characteristics of sparse connection and weight sharing and greatly reduces

the calculation of the weight parameter in the network [11]. Convolutional neural networks have achieved good results in pulmonary nodule detection. For example, Tajbakhsh N et al. used an artificial neural network and convolutional neural network to detect pulmonary nodules in CT images, and experimental results showed that the performance of the convolutional neural network was better than that of the artificial neural network [12]. Ginneken B V et al. [13] extracted three axial plane features of the detected candidate pulmonary nodules. The features of 4096 pulmonary nodules were extracted from the penultimate layer of the convolutional neural network. Finally, pulmonary nodules were classified by a support vector machine. The conclusion was that the features extracted by the convolutional neural network had great potential in detecting pulmonary nodules. Zhu W et al. [14] proposed a gradient enhancement machine with the characteristics of a 3D two-path network to detect and classify pulmonary nodules. Kang G et al. [15] proposed a 3D multiview convolutional neural network (MV-CNN) to detect pulmonary nodules, and high accuracy of the test results was obtained, showing the effectiveness of the 3D convolutional neural network. In addition to pulmonary nodule detection, the convolutional neural network is expected to play an important role in the diagnosis of more kinds of diseases. It can help doctors diagnose the patient's condition, reduce the workload, and reduce misdiagnosis and missed diagnosis.

In the convolutional neural network, the 2D U-Net architecture is mostly used in the field of medical images, as proposed by Ronneberger et al. [16]. The 2D U-Net network adopts symmetric structure, which is composed of the same number of up-sample and down-sample layers. The up-sampling can restore the details of the original image and spatial dimensions, the down-sampling can reduce the spatial dimensions of the pooling layer, and the side connection can better restore the details of the target image. During the training, the whole image can be processed by the 2D U-Net in a forward channel, which improves the training performance. However, the 2D U-Net network structure cannot fully use the 3D spatial information of images. In the scheme, the 2D network was extended to the 3D network. The structure of the 3D U-Net network was constructed, and the network input has also been changed from a 2D image to 3D image. The 3D network structure can more accurately locate the target area and obtain more comprehensive feature information, and it can effectively combine the shallow and deep features of the image, thereby improving the detection effect.

Because this paper applied a 3D network structure, the input datasets were also 3D low-dose lung CT scan images, which fully retained the image's 3D feature information. At present, TensorFlow is a new open source deep learning network platform developed by Google. The platform has released a new network interface for 3D convolution and 3D pooling, and it constructs a 3D convolutional neural network for 3D images. Therefore, this paper proposed the TensorFlow deep learning framework to construct a 3D convolutional neural network model. The model was used to extract 3D candidate nodules for training, and experimental results were obtained.

The deep convolutional neural network is a data-driven network. The training set needs to provide enough data to ensure the performance of deep learning. However, it is not easy to obtain a large number of labeled pulmonary nodule datasets in real-world settings, where the time and costs of acquisition are high. Therefore, transfer learning technology is introduced in this paper. Transfer learning is a new machine learning method that applies information and knowledge of existing fields to different but related fields. Transfer learning does not require similar fields to follow the same probability distribution, and the goal is to transfer existing knowledge and information from the source area to new areas through some technical means. In addition, it can solve the learning problem of the target domain with little or no labeled sample data [17]. Transfer learning is used to detect pulmonary nodules. First, doctors do not need to collect a large number of labeled training samples containing many different types,

which reduces the workload of the doctors. Second, training data and test data are not required to follow the same distribution, since it can use a different distribution of the training dataset to complete the learning tasks of the target domain and can effectively solve the problem of cross-domain learning [18]. Finally, transfer learning alleviates the difficulty of deep learning in introducing a small dataset of medical images because transfer learning leverages a small number of newly generated lung training sets to construct a reliable classification model through prior knowledge. Consequently, transfer learning can improve the utilization rate of lung data and the adaptability of the pulmonary nodule detection system.

In this paper, transfer learning and the deep convolutional neural network were introduced to detect pulmonary nodules. First, the lung parenchyma area was extracted. Then, the initial network model was used to extract the image features of pulmonary nodules. Finally, transfer learning fine-tuned the network model and used the pretrained network model to train the dataset. After testing and validating the dataset, the pulmonary nodules were detected. The five main contributions of this paper are listed as follows.

1) Parenchyma segmentation is the first step of the detection of pulmonary nodules. It is also a prerequisite for accurate detection of pulmonary nodules. In this paper, the adaptive threshold method, morphological grayscale reconstruction method, grayscale integral projection method and rolling ball method were comprehensively applied to accurately and effectively segment lung parenchyma. The approach lays a good foundation for the follow-up detection of pulmonary nodules.

2) TensorFlow is selected as the basic framework of the convolutional neural network, which simplifies the code structure of the network and improves the efficiency of the deep learning algorithm. The construction of the 3D convolutional neural network, training and algorithm evaluation for the lung CT image are realized. The original TensorFlow framework was also improved by the expansion of the local receptive field and extraction of more features by using different convolution kernels. The max-ave pooling method was adopted. It could improve the convergence speed of the network and retain more characteristic information.

3) This paper proposed a network structure of feature extraction on three different scales that could obtain more features of pulmonary nodules and improve the accuracy of detecting small nodules. Dense connection blocks were added to the network structure. Under the premise of ensuring the maximum information transmission between network layers, all layers were directly connected to prevent overfitting.

4) Transfer Learning was introduced into the detection of pulmonary nodules, and the image features were extracted by the multiscale feature structure the 3D U-Net model. It can not only make up for the overfitting problem caused by a smaller dataset but also save considerable time for the doctor's task of manually marking samples. Therefore, it could help doctors to diagnose lung cancer more quickly and accurately.

5) A variety of parameters were constantly adjusted that affect the performance of the 3D convolutional neural network model, and the experimental results were compared and analyzed to obtain the best performing model. Furthermore, the experiment verified the accuracy of different sizes of nodules. Finally, the proposed scheme was compared with the existing detection algorithm, and the experiment verified the feasibility of the proposed scheme.

The rest of this paper is organized as follows. The dataset used in this research for detection of pulmonary nodules is introduced in Section 3. The proposed scheme is described in Section

4. The experimental results are presented in Section 5. The conclusion is drawn in Section 6. The advantages and disadvantages of this scheme and the key directions of future research are presented in Section 7.

## Pulmonary nodules detection dataset

In this paper, two medical datasets were applied to verify the feasibility of this scheme, as shown below.

TIANCHI17 dataset: In early 2017, a dataset of 800 low-dose lung CT images (mhd format) were provided in the preliminary of Aliyun Tianchi big data competition. The original images are 3D images. The 3D images are composed of 2D images of multiple axial slices. The number of 2D image slices varies with different machine scans, scanning thicknesses and patients. The dataset included 1000 patients with pulmonary nodules, where lung nodule sizes of 5 mm-10 mm accounted for 50% and 10 mm-30 mm accounted for 50%. All the pulmonary nodules were identified by three experts. The layer thicknesses of all scanned images are less than 2 mm [19].

The Lung Nodule Analysis 16 (LUNA16) dataset [20]: This dataset of pulmonary nodule detection was released in 2016. LUNA16 data comes from a larger set of the lung image database consortium and image database resource initiative, the LIDC-IDRI dataset [21]. The dataset contains 1018 CT images of 1010 patients, each CT image contains a tag file in XML format, and the dataset comes from seven different academic institutions. Because the scanner and parameters are different, the 1018 sets of images are unevenly distributed. The LUNA16 dataset excluded slice thicknesses greater than 3 mm; at the same time, the inconsistent space and missing parts were also removed. In total, 888 CT scans were included, which constituted the LUNA16 dataset. In the experiment, all nodule sizes were greater than or equal to 3 mm, and nodules had been labeled by at least 3–4 radiologists.

## Methods

### Overview

In this section, the proposed computer-aided detection scheme is described in detail. The proposed scheme consists of two major steps: lung segmentation and pulmonary nodule detection. The detailed process of lung segmentation is shown in Fig 1, and the detection process of pulmonary nodules is shown in Fig 2.

Regarding the aspect of lung segmentation, the scheme proposed the comprehensive application of multiple methods to segment the lung region. Step 1: The thoracic cavities region was roughly segmented. Step 2: The lung trachea were reconstructed and the main trachea and bronchus adhering to the lung parenchyma were removed. Step 3: The adhesion of left and right lungs was identified, and left and right lungs were separated. Step 4: the boundary of the lung area was repaired and completed the segmentation of lung parenchyma. The method can not only accurately segment the lung parenchyma but also well separate the lung parenchyma from the bone, muscle and other tissues inside the chest cavity.

From the aspect of pulmonary nodule detection, the scheme proposed an algorithm for detecting pulmonary nodules that used multiscale feature structures in the 3D U-Net convolutional neural network. The 3D network structure can make full use of 3D spatial information and can effectively combine the shallow and deep features of CT images to extract richer feature information. The irregular shape of pulmonary nodules and the low detection accuracy of small pulmonary nodules were considered. In this paper, feature extraction on three different scales was proposed, such that the features of the lower layer, the middle layer and the top

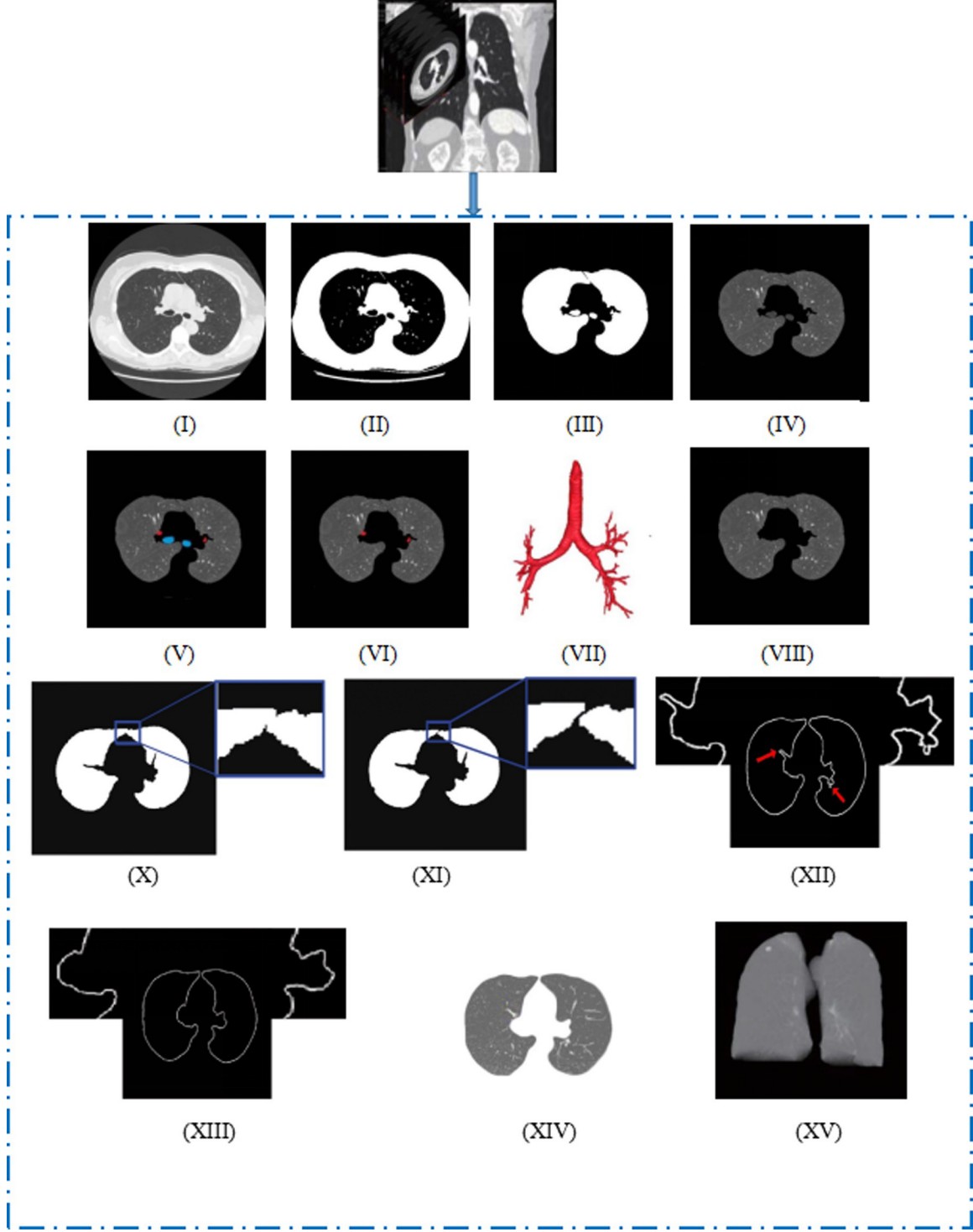

**Fig 1. Lung segmentation process diagram.**

layer are fused to obtain the features of pulmonary nodules as much as possible. This method improved the accuracy of detecting irregular and small pulmonary nodules.

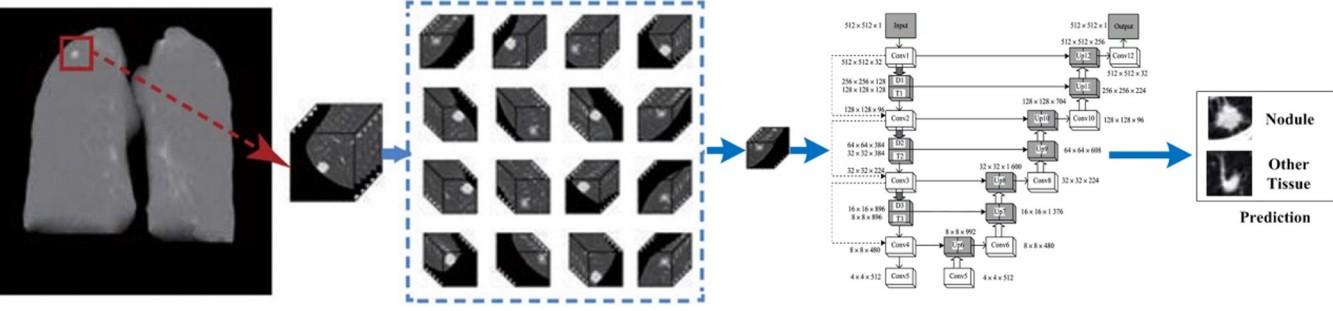

**Fig 2.**

## Lung segmentation

Because pulmonary nodules exist only within the lung parenchyma, the lung parenchyma needs to be segmented such that the interference outside is reduced. In this paper, a combination of multiple methods was used to accurately segment the lung parenchyma. The specific segmentation steps are as follows.

Step 1: The lung parenchyma was roughly segmented. (1) The original chest image was selected from the middle layer of lung CT, as shown in Fig 1(I). (2) The adaptive threshold algorithm was used to binarize the image [22], as shown in Fig 1(II). (3) The maximum connectivity region labeling method was used to remove the tray of the background and image below, as shown in Fig 1(III). (4) The human skeleton, muscle and other tissues were removed by the seed filling method, and the rough segmentation results of lung parenchyma after masking were obtained, as shown in Fig 1(IV).

Step 2: Removal of lung trachea and alveolar tissue. After the image, which contained a large amount of lung trachea and a small amount of alveolar tissue in the trachea wall of lung, was roughly segmented, the lung trachea needed to be removed to obtain an accurate lung region. This paper proposed a method of lung tracheal segmentation based on optimal threshold growth and morphological gray reconstruction. In the rough segmentation result Fig 1(V), there are two lung main trachea (blue mark) and two bronchi (red mark). The steps of removing lung trachea and other tissues are as follows: (1) The main trachea was removed by the optimal threshold growth algorithm, and the diagram is shown in Fig 1(VI), but the bronchus still existed. (2) To remove the bronchus, first, the morphological closed operation was used to manage lung parenchymal images. Then, images were 3D gray reconstructed. The finer trachea trees were constructed, and the method enriched the details of the bronchus tree and improved the segmentation accuracy of the bronchus. The 3D diagram is shown in Fig 1(VII). Finally, the reconstructed image subtracted the original lung parenchyma image to obtain the bronchi region and the bronchi were removed. (3) The alveolar tissues around the lung trachea were removed by the morphological operation to obtain a finely segmented image of the lung parenchyma, as shown in Fig 1(VIII).

Step 3: The adhesion of left and right lungs was identified, and left and right lungs were separated. Due to the presence of the local volume effect, the similar two lung regions easily stick together, which will affect the subsequent repair of the lung area and the detection of pulmonary nodules; therefore, the two lung regions need to be separated [22]. Steps of separation: (1) Judging the fusion of two lung regions, when the left and right lungs were fused, there would be a connected area larger than the sum of the two lung areas in the whole lung area, and the connected area would be much larger than the second connected area. Based on the above theory, this paper used the component analysis method to determine whether the left and right

lungs were fused;, the basic idea was that if the difference of the first and second largest areas was greater than a threshold (T), the right and left lung adhesion was fused. After several tests, it was found that when T = 4, it can accurately determine right and left lung adhesion, as shown in Fig 1(IX). (2) The vertical integration projection method was used to determine the location of right and left lung separation. (3) The segmentation point was obtained by line scan, where linear interpolation of black pixels was employed on the dividing line composed of dividing points, and the segmentation line was broken to achieve the separation of the left and right lungs, as shown in the segmentation results in Fig 1(X).

Step 4: The boundary of lung area was repaired. After lung parenchyma was divided, pulmonary nodules and blood vessels that adhered to the lung wall were easy to be missed in the lung area, and the boundary of the lung area was concave. To ensure the detection accuracy of pulmonary nodules, it is necessary to repair the lung boundary and fill in the depression of the lung boundary. In this paper, the left and right lung areas were repaired by the rolling ball method [23]. The rolling ball method used the ball to roll along the border of lung, when two or more points of the ball touched the depression of the lung area, the rolling track of the ball replaced the depression boundary to realize the repair of the lung boundary. After many tests and comprehensive consideration, the radius of the ball r = 12.5 mm was found to better repair the depression boundary of the lung area. Fig 1(XI) is the image before repair; Fig 1(XII) is the image after repair; Fig 1(XIII) is the final segmentation image of the lung parenchyma, while Fig 1(XIV) is a 3D reconstruction of the lung parenchyma.

## 3D U-net convolutional neural network structure

Fig 3 is the structure of the 3D U-Net convolutional neural network. The network structure includes the convolution layer, pooling layer, activation function, up-convolution layer and concat layer [24]. To reduce the computation of training, this paper adopted the fixed-size 3D image block input network structure. Image features were constantly extracted to judge the probability that each individual element attributed to the nodules, and the predicted results were output.

The convolutional layer extracted image features by local connections between layers and convolution operations of different convolution kernels, the extracted results were mapped by

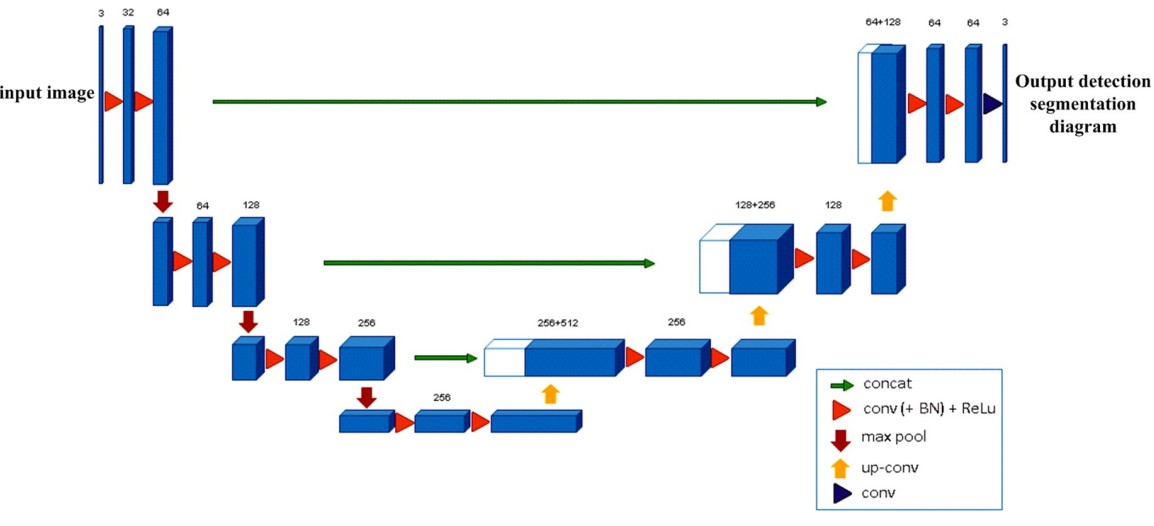

**Fig 3.**

a nonlinear activation function, and different feature graphs were obtained. Activation functions include sigmoid, tanh, softsign and ReLU, etc., After experimental comparison, ReLU was selected in this paper. ReLU can accelerate the convergence speed of the network, overcome gradient dispersion and reduce the training time in the back-propagation [25].

The pooling layer mainly extracts and combines the similar features in the convolutional layer, reduces the output characteristics of the convolutional layer and prevents overfitting. Pooling methods include max pooling, average pooling and random pooling. After experimental comparison, the max pooling method was selected, The method could retain the main features that reduce the total number of parameters, reduce the calculation time and improve the generalization ability of the model [26].

The deconvolution layer was the inverse operation of the convolution layer; after a series of convolution and pooling operations, the resolution of the feature graph was reduced. After the deconvolution operation, the size of the feature graph was restored.

The cascade layer merges the output layer of the first layer with the input layer of the second, It's easy to lose the boundary pixel information in the convolution operation, The cascade layer can obtain more feature information of the target area through the combination of the high-level feature and the low-level feature, which can achieve the purpose of better repairing the target details.

The cascading layer was the combination of the output layer of the first layer and the input layer of the second layer, which could increase the number of features and the performance of the network and could better repair the details of the target.

The last convolutional layer was the output layer that extracted the high-level semantic features of the image and added the ReLU activation function to output the probability graphs of different categories [27]. Since the cross-cost function was used, the background marked as zero would not be calculated in the loss, which could improve the convergence speed.

The 3D U-Net network model is a symmetrical structure; downsampling can reduce the space dimension of the pooling layer, and upsampling can repair the feature details. However, the 3D U-Net network structure cannot make full use of low-level information and consider more high-level features; low-level information is rarely reused, and the target area is not analyzed from the multiscale perspective. At the same time, the features' continuities are rarely considered, and the relationship between the image itself and the target are ignored. It leads to the problem of unclear location and large deviation in the detection area. Finally, the detection result of pulmonary nodules is not ideal.

## Multiscale feature 3D U-net convolutional neural network structure

To accurately detect pulmonary nodules, based on the characteristics of pulmonary nodules, a multiscale feature detection algorithm was proposed in the 3D U-Net network structure. The algorithm is as follows: 1) To ensure the continuity between features, dense networks were added to the down-sampled 3D U-Net networks. In this way, features from low to high of different scales were constructed, and features of different scales could detect targets of different scales. 2) Considering the small shape and few semantic features of pulmonary nodules, the expanded convolution was added to the downsample, as far as possible, and the low-level features were reused to provide pixel-level features for detection tasks. 3) In the up-sampled the 3D U-Net network, the continuous upsample was adopted instead of the once upsample approach. The semantic feature of pulmonary nodules could be maximally utilized to obtain the low-level features of small pulmonary nodules and reduce the loss of features.

The network structure is shown in Fig 4. The down-sampled 3D U-Net network included four convolution layers, four pooling layers, three dense blocks and three extended

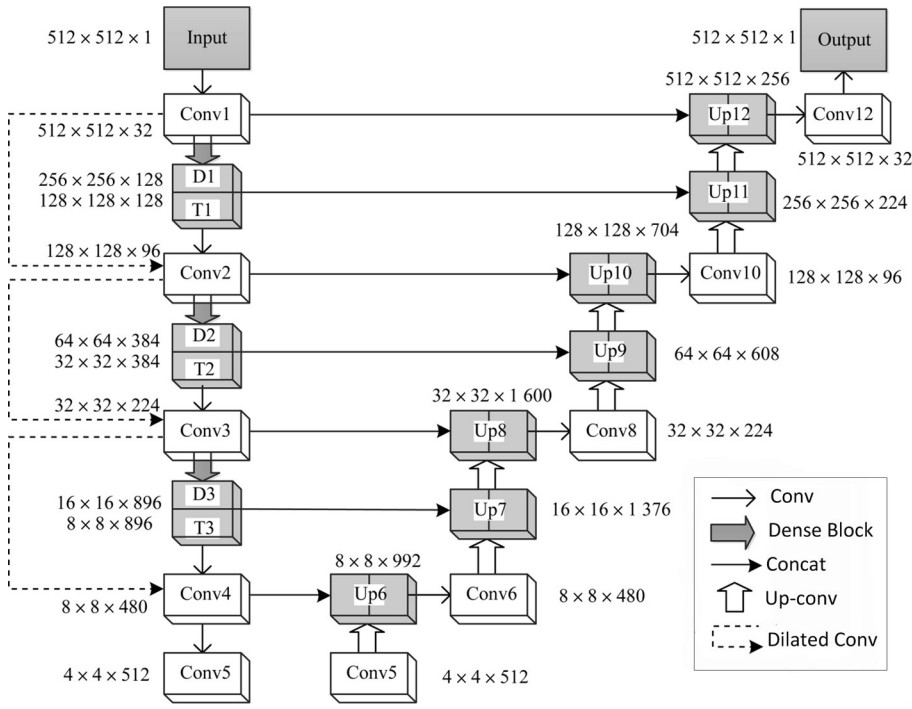

**Fig 4.**

convolution layers. It was mainly used to extract useful features in images. The up-sampled network included five up-conv blocks and seven characteristic patterns after the up-conv layer.

The Up-conv layer could not only increase the resolution of the characteristic pattern but also learn the context information. The output layer consists of two 1×1 convolution layers. The sigmoid activation function was used to output the probability of the positive samples (pulmonary nodules). This paper introduces the net structure in detail.

In Fig 4, $D_n$ (n = 1,2,3) represents the Dense Block in the Dense network, $T_n$ (n = 1,2,3) represents the Transition Layer, and Up n (n = 6,7, . . .,12) represents the characteristic pattern that is obtained by the combination of up sample and the side connection. For the input 512×512×1 CT image, the original image is converted to 512×512×32 Conv1 using the 3×3 convolution kernel. Conv1 performs two operations. Operation1: Conv1 was pooled, and it was connected to the Dense Block and the Transition Layer, represented by D1 and T1 in Fig 4. The growth rate of dense blocks is 3. The operation obtained 256×256×128 feature layers of D1 and 128×128×128 feature layers of T1. T1 was convolved by 3×3 to obtain the characteristic layer Conv2 of 128×128. Operation2: After Conv1 was pooled, the 3×3 expansion convolution was added, and obtained the 128×128×96 feature layer Conv2. Operation 2 used the extension convolution to reduce the loss of underlying information. A few feature losses are allowed in image feature extraction, so operation1 uses the maximum pooling method.

Feature layer Conv2 also requires a two-step operation, the join dense block operation and the expansion convolution downsample operation. After the Conv2 was connected to the dense block, the feature layer Conv3 of 128×128×128 was obtained through the 3×3 convolution operation. Then, feature layer Conv3 of 96×128×128 was obtained through the down sample operation of the expanding convolution. The two operations were combined to obtain Conv3 of 224 feature graphs. As a series of Conv3 operations, 480 characteristic patterns with

8×8 feature layers were obtained. After the above operation, the multiscale feature structure is formed, and feature layers Conv2, Conv3, and Conv4 constitute the features of low, medium and high scales.

In the process of upsampling, feature layer Conv5 was deconvolved to obtain the 512×8×8 feature layer Up6. At the same time, feature fusion was conducted between Conv4 and Up6 to obtain the 8×8×992 feature layer Up6. Then, the feature layer Conv6 of 8×8×480 was obtained by the convolution operation. After two upsamples, the feature layer Up8 of 32×32×1600 was obtained. Conv8 of the feature layer was obtained through the Up8 convolutional layer as well, and then, the Conv8 feature layer was made the same size and dimension as Conv3. Finally, two dense blocks were added, continuously up-sampling twice, and the feature layer Up12 with a size of 512×512 and a number of 256 was obtained. At this time, the scale of the characteristic pattern was consistent with the original graph. The output layer was two convolutional layers, which output the same size and channel number as the original graph.

## Multiscale feature structure

In the process of pulmonary nodule detection, it was necessary to obtain more features of pulmonary nodules that improve the accuracy of detection. The 3D U-Net network structure only uses the top-level semantic features when the features are continuously extracted in the down-sample, and the middle and lower features of pulmonary nodules are ignored, incurring a characteristic loss. In this paper, the multiscale feature structure model was constructed by using the multilayer connection of dense blocks and the expansion convolution, as shown in Fig 5. The low-level feature graph was fused with the high-level feature graph. Finally, a feature graph with high-level semantics and rich location information was obtained.

## Densely connected network structure

When the number of layers of the neural network deepens, the gradient vanishing phenomenon appears. One approach to the problem was to shorten the connection distance between the initial input and the loss function. Dense connections make each layer of the network directly connected to inputs and losses. The phenomena of gradient dissipation and regularization are alleviated, and the overfitting of the network is restrained [28]. Fig 6 shows the dense

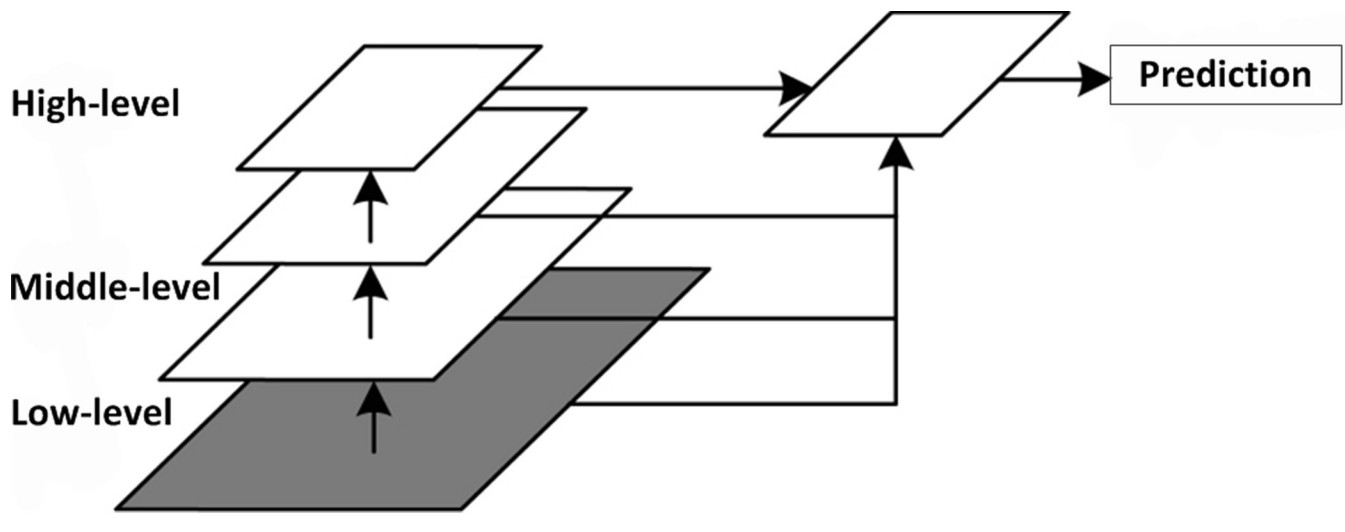

**Fig 5.**

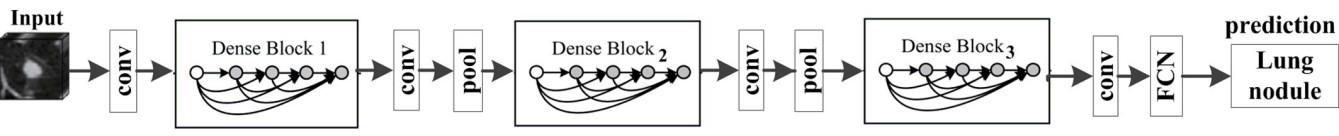

**Fig 6.**

network structure, and the design process is as follows: The first step was to input the feature image and execute a convolution of 1*1 on the image and fusion features of all channels in the feature graph, such that dimension reduction could be performed. In the second step, in Fig 6 as an example, the fixed growth rate of dense blocks is set as 3. In the third step, the convolution operation was used to reduce the dimension of the characteristic pattern, and then, the pooling operation was added to reduce the network parameters. The fourth step was to perform the convolution operation of 1*1 that reduces the dimension and outputs the prediction result. Dense connected networks have the ability curtail the phenomenon of gradient dissipation and reduce the size of the network model, given that network models can reuse image features. Consequently, a large number of features can be generated by fewer convolution kernels, thus reducing the number of network parameters.

## Transfer learning fine-tuning network structure

The structure of the 3D U-Net convolutional neural network needs a large number of marked training samples to complete the training task, and the training samples and test samples are required to follow the same distribution. However, it consumes much human and material resources to annotate medical image data. Moreover, the accuracy cannot be guaranteed, which it makes difficult to obtain reasonably labeled training data. In this paper, the transfer learning method was selected to solve the above problems. Transfer learning can solve the learning problem of a small amount of labeled data or even no labeled data in a new field through original knowledge. Transfer learning can use prior knowledge to solve problems of different but related fields. It can effectively solve the problem of the lack of training data and meet the needs of small medical image datasets. Therefore, this paper introduces the idea of transfer learning to fine-tune the parameters of different layers of the 3D U-Net network model that optimizes the training process.

This paper mainly adopts the transfer learning method based on weight to adjust the network model; the process is shown in Fig 7. First, data enhancement, can effectively reduce the

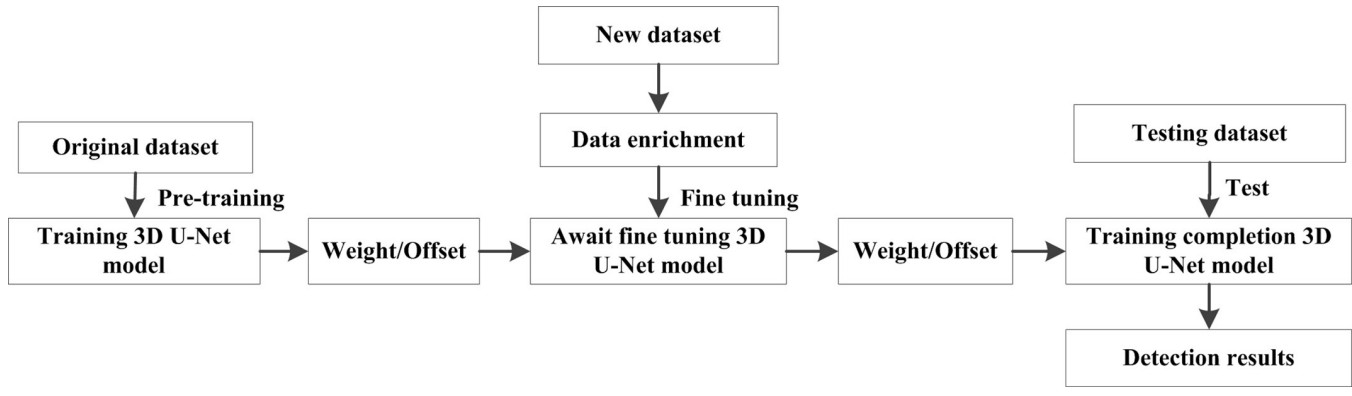

**Fig 7.**

overfitting and improve the training efficiency of the model [29]. The data enhancement adopts horizontal, vertical, rotate 90˚, 180˚ and 270˚ [30] approaches. The number of training samples after enhancement was 6 times that of the original. The enhanced dataset of different sizes of pulmonary nodules is fed into the 3D U-Net model for training. Then, the initial weight is obtained. After the training of the original dataset, the first four convolutional layers of the network mainly learn the underlying feature information such as the points and edges of images. The similarity of this information can be migrated in the new dataset to process the image. The trained weights can be taken as the initial weights of the feature extraction layer in the new dataset. Then, the stochastic gradient descent optimization method (SGD) is used to fine-tune the weight, which better reflects the characteristics of pulmonary nodule dataset. Transfer learning methods generally fine-tune the last few layers of the network. To obtain more image features, the previous layers are also fine-tuned. The network layer parameters are updated from the last layer. The required performance is achieved to obtain the best model. Finally, the test dataset is input and the model's detection effect is validated by the five-fold cross-validation method.

## Network training

In the data training stage, first, the TIANCHI 2017 and LUNA 16 datasets are preprocessed. Then, the two dataset are extracted to train the 3D U-Net model. After fine-tuning the model parameters, the TIANCHI 2017 dataset is used to train the adjusted model. Finally, the performance of the network model is tested with the remaining dataset.

The preprocessing was mainly to standardize the resolution of data. Because of the difference of machines or parameter settings, the resolution of CT images varies greatly, Therefore, it was necessary to standardize the resolution of medical images. For this experiment, the size of $1.0 \times 1.0 \times 1.0$ mm$^3$ was selected for standardization, The gray mean and standard deviation of the training data were calculated, and the mean and standard deviation are used to standardize the image's gray value.

This experiment was based on the TensorFlow deep learning framework of the Ubuntu 14.04 system. A total of 5,328 enhanced images of pulmonary nodules with expert markers were input into the 3D U-Net network model to detect pulmonary nodules through transfer learning. The adaptive moment estimation (Adam) [31] optimization algorithm was used to estimate the cross-entropy loss of each pixel between the output image and the expert marker image. Adam is an adaptive learning rate algorithm, and the formula is:

$$\left. \begin{aligned} m_t &= \mu * m_{t-1} + (1-\mu)^* g_t \\ n_t &= \upsilon * n_{t-1} + (1-\upsilon) * g_t^2 \end{aligned} \right\} \tag{1}$$

$$\left. \begin{aligned} m_t{}^\wedge &= \frac{m_t}{1-\mu^t} \\ n_t{}^\wedge &= \frac{n_t}{1-\upsilon^t} \end{aligned} \right\} \tag{2}$$

$$\Delta\theta_t = -\frac{\eta}{\sqrt{n_t{}^\wedge} + \varepsilon} * m_t{}^\wedge \tag{3}$$

The parameters are described as follows: $g_t$ is the update gradient of $t$ times the training parameter; $m_t$ is the first-order (mean) moment estimation, and $n_t$ is the second-order (variance) moment estimation; $\mu$ is the momentum factor of $m_t$, $\upsilon$ is the momentum factor of $n_t$, $\mu = 0.9$, $\upsilon = 0.9999$; $m_t^\wedge$ is the correction of $m_t$, $n_t^\wedge$ is the correction of $n_t$; $\mu_t$ is t power $\mu$, and $\upsilon_t$ is t power $\upsilon$.

Formula (2) was used to solve the problem of the deviation of parameters toward zero at the beginning of training. $\Delta\theta_t$ represents the update quantity of the parameter; $\eta$ represents the initial learning rate, $\eta = 0.01$, and $\varepsilon$ represents the weight attenuation coefficient, $\varepsilon = 1 \times 10^{-8}$.

After bias correction, the Adam algorithm calculates the adaptive learning rate for each parameter, and each update learning rate has a certain range to make the parameters more stable. The weight after 3D U-Net training was taken as the initial weight. To reduce the overfitting, the parameter dropout was set as 0.5. Dropout means temporarily deleting a certain probability neuron at each reverse update of the neuron parameters, but keeping the neuron parameter for the next update [32]. This parameter can reduce the dependence between adjacent neurons, avoiding neural networks that are overly sensitive to some features that enhance the robustness of the network.

The parameters for evaluating the network training are: training accuracy (*TrainAcc*) and overfitting ratio (*OverRatio*).

$$TrainAcc = \frac{Train\ ImageCurrently}{Training\ Images} \qquad (4)$$

*TrainImageCurrently* represents the number of correctly detected images, and *Training Images* represents the total number of training set images;

$$OverRatio = \frac{TrainAcc}{ValAcc} \qquad (5)$$

*TrainAcc* represents the accuracy of the training set, *ValAcc* represents the accuracy of the validation set.

The LUNA16 dataset was used for training. A total of 50,000 training sessions were conducted, and a model was saved after each 10,000 training sessions. Training was stopped in advance when there was overfitting. The images of the validation set were experimentally verified, and the results are shown in Table 1. The training results showed that the model with 30,000 training sessions had the best results. Although 40,000 and 50,000 models have been trained more times, the overfitting phenomenon occurred. Therefore, this paper selected 30,000 training results as the optimal training results of this experiment. As the number of training increases, the loss decreases gradually, and the loss value was recorded once for every 20 training sessions. To display more smoothly, by recording loss values 30 times, the average value was obtained, as shown in Fig 8. Fig 8 shows that the loss value was only 0.19 at 30,000 training sessions and the initial loss value was 0.62. Experiments verify the feasibility of transfer learning.

During the training process, the stochastic gradient descent algorithm was adopted. The data for each round of training was approximately 88 pictures. The learning rate (lr) was 0.00075, 0.001 and 0.005, and the dropout was 0.5 for training. The training results are shown in Fig 9.

After 60 rounds of training, Fig 9 shows the training results with different parameters. Fig 9 (A), 9(C) and 9(E) show the change of the error rate of the training set and test set in network training. Fig 9(B), 9(D) and 9(F) show the change of the accuracy of the training set and test

**Table 1. Comparison of results of different training times.**

| Evaluation index | Training times(n) | | | | |
|---|---|---|---|---|---|
| | **10000** | **20000** | **30000** | **40000** | **50000** |
| Training accuracy(%) | 72.15 | 85.32 | 93.48 | 93.45 | 93.49 |
| Overfitting rate | 1.3246 | 1.221 | 1.071 | 1.145 | 1.123 |

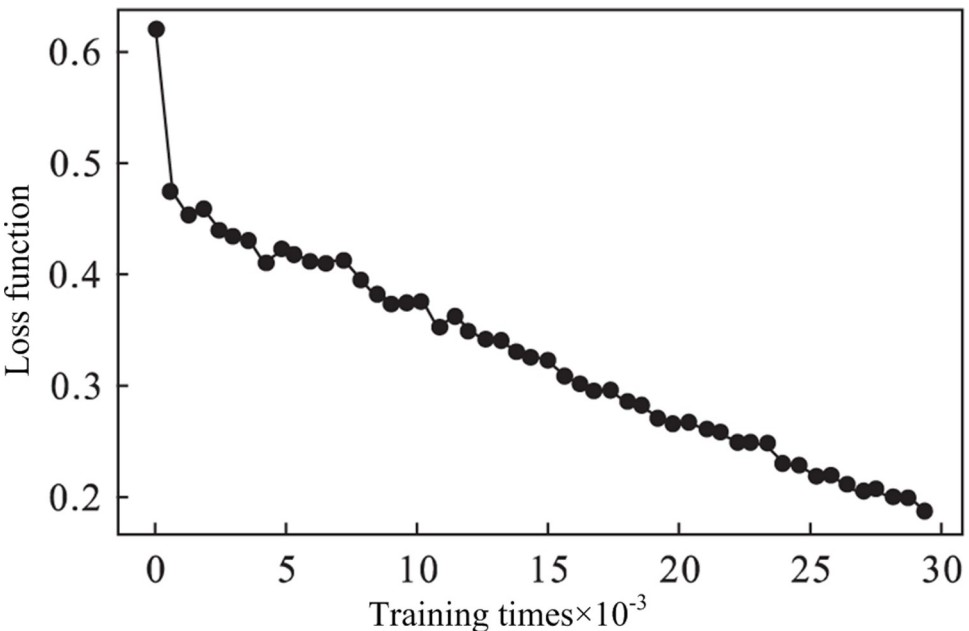

**Fig 8.**

set. The horizontal ordinate shows the number of training times with all samples, and the ordinate shows the error rate and accuracy rate. From the results in the Figure, it can be concluded

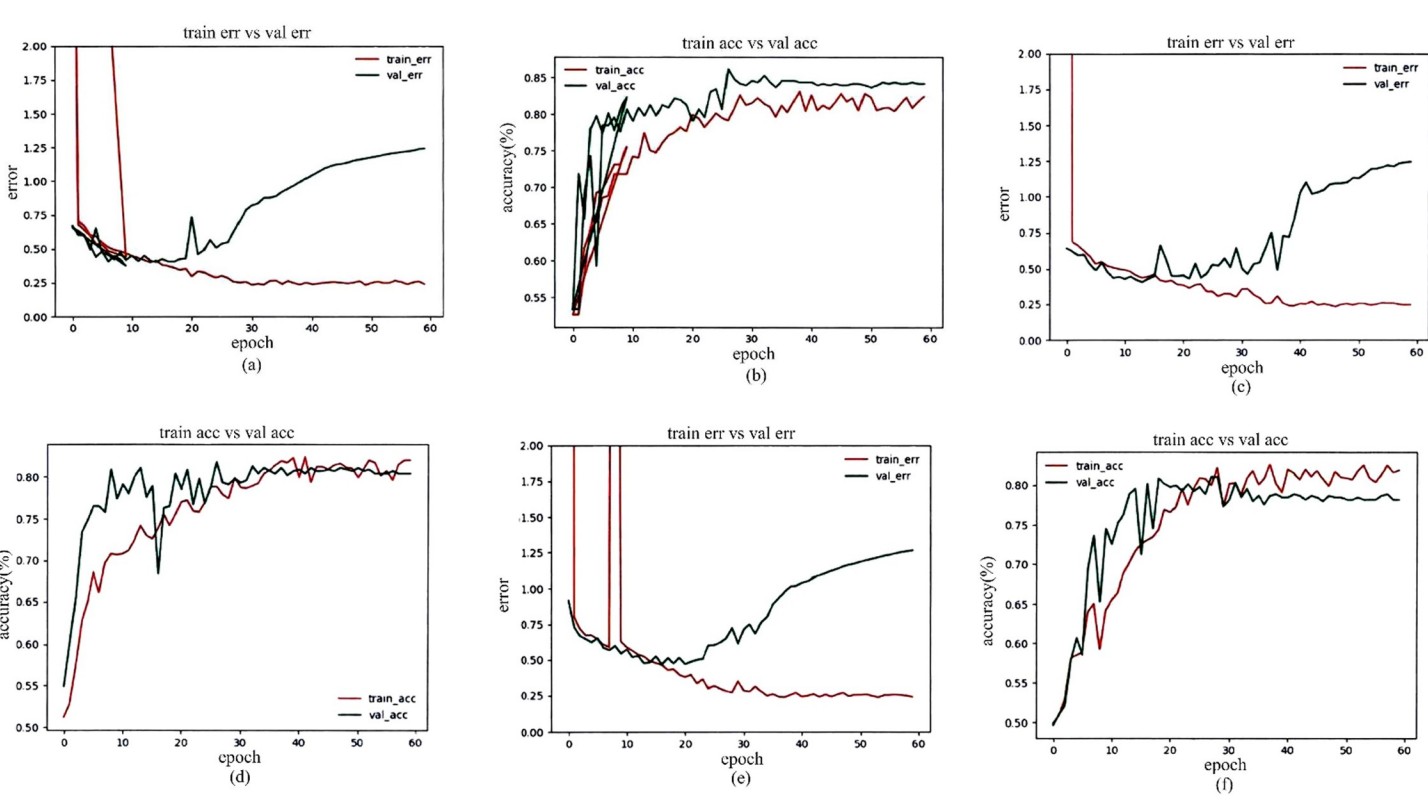

**Fig 9.**

that the selected Ir and dropout value could obtain a lower error rate and higher accuracy. When $Ir = 0.00075$, *dropout* = 0.5 (Fig 9(B)), the training result had the highest accuracy. The experimental results showed that the lower learning rate and the smaller updating range of parameters could obtain a higher training accuracy.

## Experimental results and analysis

### Dataset description

To verify that the proposed multiscale feature structure method can detect pulmonary nodules of different sizes, pulmonary nodules were classified into small nodules, middle nodules and large nodules. Small nodules had a diameter less than 5 mm, middle nodules had a diameter between 5 mm and 15 mm, and large nodules had a diameter greater than 15 mm. For each candidate nodule, the size of 40 pixel×40 pixel×26 pixel 3D image blocks was selected to completely include the nodule, and pulmonary nodules images of different sizes are shown in Fig 10. LUNA16 is a historical dataset, which was divided into 10 subsets, and the 3D U-Net network model is trained on each subset. The number of each subset and sections of pulmonary nodules in the subset are equally distributed, as shown in Table 2. Datasets were trained by the five-fold cross-validation method. Then, 600 images were selected in the TIANCHI 2017 dataset as the new dataset that continued to train the model after fine-tuning parameters, and the algorithm was validated with 200 validation images. The pulmonary nodules' size distribution of the two datasets is shown in Fig 11. The distribution trends of the two lung nodule datasets were similar. This result shows that there was little difference between the two datasets, and the transfer learning method could be applied to train the datasets.

### Experimental environment

The TensorFlow deep learning framework and Python language environment were implemented on a Ubuntu 14.04 operating system to complete the experiment [33]. Detailed hardware and software environment are shown in Table 3.

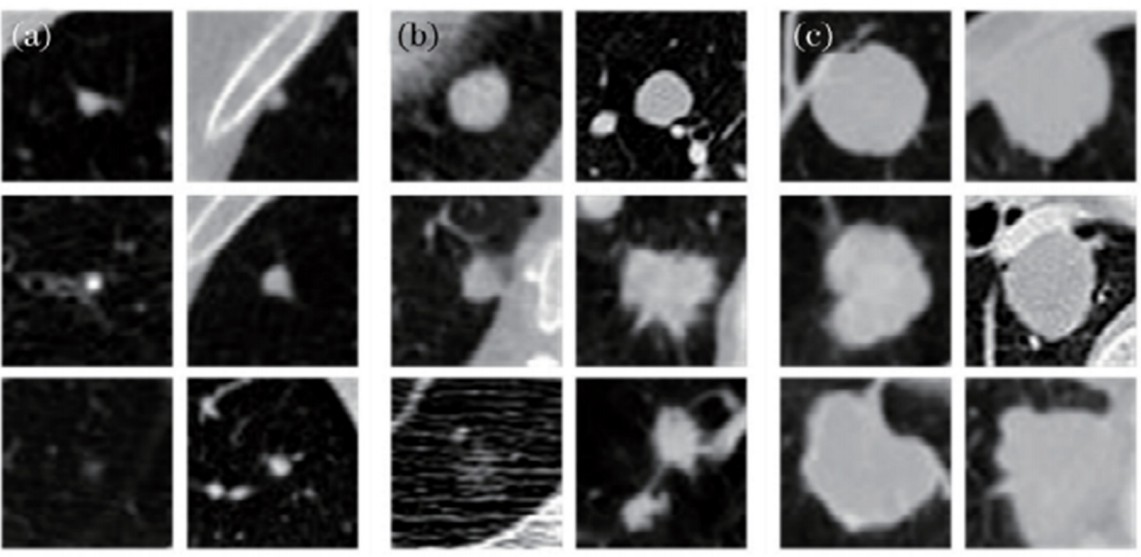

**Fig 10.**

**Table 2. The number of each subset and sections of pulmonary nodules in subset.**

| Subset | 0 | 1 | 2 | 3 | 4 | 5 | 6 | 7 | 8 | 9 |
|---|---|---|---|---|---|---|---|---|---|---|
| Total number of samples(n) | 89 | 89 | 89 | 89 | 89 | 89 | 89 | 89 | 88 | 88 |
| Number of samples containing nodules(num) | 67 | 61 | 56 | 65 | 62 | 54 | 63 | 54 | 60 | 59 |
| Mean of section containing nodules(avg) | 13.2 | 16.6 | 18.5 | 14.4 | 15.6 | 14.7 | 14.8 | 14.1 | 18.2 | 13.8 |
| Largest number(max) | 35 | 54 | 60 | 49 | 70 | 70 | 70 | 46 | 59 | 46 |
| Minimum number(min) | 3 | 3 | 3 | 3 | 3 | 3 | 4 | 4 | 3 | 4 |
| Total number(sum) | 885 | 1016 | 1036 | 938 | 972 | 798 | 935 | 763 | 1094 | 819 |

## Analysis of factors affecting the performance of network models

The experiment aimed to verify the feasibility of the selected four parameters based on the network model in this paper, which are the convolution layer number, pooling method, activation function and optimization mode. The influence of each parameter on the performance of the network model is analyzed.

## Different convolution layers

The number of convolutional layers was gradually increased based on the multiscale feature of the 3D U-Net convolutional neural network model. Through experiments consisting of twenty iterations, the experimental results are shown in Table 4.

Table 4 shows that the accuracy of the network structure detection increases with the increase in convolutional layers, and the loss value gradually decreases and tends to be stable. The performance of the network reaches the best when the number of convolutional layers was four with other parameters unaltered, and the accuracy of more than six layers tends to be stable. Therefore, the number of convolutional layers was selected as four during the downsample method in this paper.

## Selection of pooling methods

When the image was downsampled through the pooling layer, pooling operations can be used to tailor images on the basis of preserving image features, such that the computation is reduced

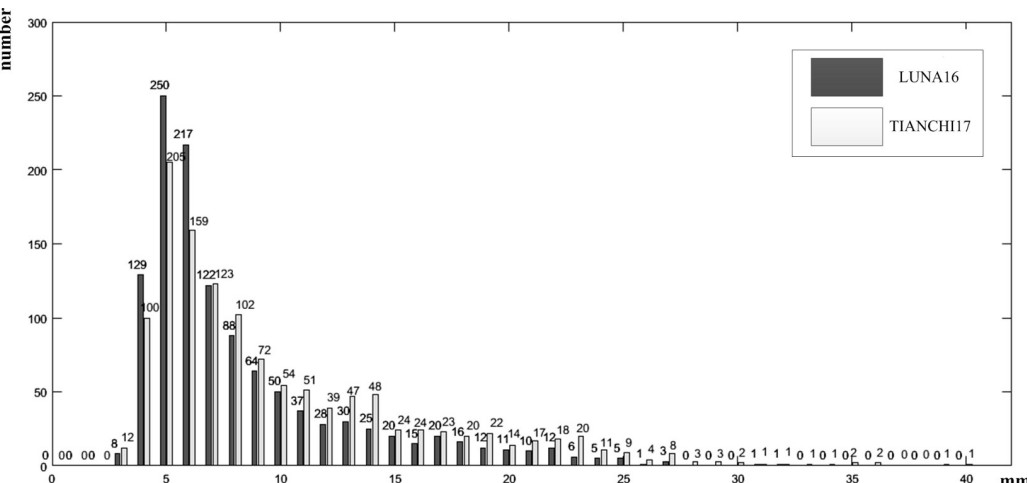

**Fig 11. Pulmonary nodules distribution of radius in CT images with diameter of 3–30 mm.**

Table 3. Hardware and software environment configurations.

| Experimental environment | Configuration instructions | |
|---|---|---|
| Hardware environment | CPU | Intel(R) Core(TM) i7-6700 CPU 4GHz |
| | GPU | Nvidia GTX 750, 4GB |
| | Memory | 8GB |
| Software environment | Operating System | Ubuntu 14.04, 64 bit |
| | Programming environment | Tensorflow deep learning framework and Python language |

and the convergence rate of the network is accelerated. The commonly used pooling methods include max pooling, mean pooling, stochastic pooling, etc. [34]. The experiment was carried out with other parameters unchanged, and the experimental results are shown in Table 5.

Selecting the max pooling method can retain more image feature information and obtain higher detection accuracy, as seen from Table 5. The average pooling or random pooling method was selected to extract the average or random value of the region to represent the output pixel value, which would lose the main information and reduce the detection accuracy.

## Selection of activation function

The activation function has a great influence on the training ability of the network model. An appropriate activation function was selected, which can effectively reduce gradient descent, accelerate network convergence, and improve detection accuracy. The common activation functions are the sigmoid function, tanh function, ReLu function, ELU function, ReLu6 function, and softplus function [35]. Different activation functions were selected to conduct experiments on the network model, and the experimental results are shown in Table 6.

Comparative analysis of the results in Table 6 reveals the following. In the training process of the convolutional neural network, gradient dissipation very easily occurred when the back-propagation algorithm was adopted, which leads to an increased loss value and low accuracy. The tanh function was similar to the sigmoid function, as they both have the gradient dissipation problem. However, the tanh function's output was zero mean, and it has a greater tolerance ability and higher accuracy than the sigmoid function. The ELU function has the highest detection accuracy, but its loss value was greater than the ReLu function. ReLu is an unsaturated linearly modified piecewise function, and no gradient dissipation problem occurred. After the back-propagation algorithm was adopted, the convergence was very fast, so the ReLu function has more advantages than other functions. The ReLu function has higher accuracy and the lowest loss value from the results in Table 6. Therefore, the ReLu function was selected in this paper.

## Optimizer selection

Loss or cost functions are often used as evaluation criteria in evaluating the application performance of networks. In this paper, cross-entropy was selected as the cost function in the

Table 4. The performance comparison of different convolutional layers.

| network model | evaluation index | convolutional layers | | | | |
|---|---|---|---|---|---|---|
| | | 2 | 3 | 4 | 5 | 6 |
| multi-scale feature | accuracy rate(%) | 74.992 | 83.333 | 96.679 | 96.675 | 96.673 |
| 3D U-Net | loss value | 0.471801 | 0.473603 | 0.456618 | 0.481990 | 0.450826 |

**Table 5. The performance comparison of different pooling methods.**

| pooling methods | accuracy rate (%) | loss value |
|---|---|---|
| Max Pooling | 92.459 | 0.413981 |
| Mean Pooling | 85.214 | 0.469739 |
| Stochastic Pooling | 67.548 | 0.493351 |

network model. The cost function can effectively reduce when a reasonable optimizer is selected in the training process of network model. Four optimizers are commonly used to optimize the network model in the TensorFlow platform: 1) tf. train. Adadelta Optimizer, 2) tf. train. GradientDescentOptimizer, 3) tf. train. Ada-gradOptimizer, and 4) tf. train. Adam Optimizer [36]. The experiment was carried out with other parameters unchanged, and different optimizers were selected for the experiment. The results are shown in Table 7.

From the comparison and analysis of the results in Table 7, it can be concluded that the model has the highest detection accuracy and the lowest loss when selecting the tf.train.Adamoptimizer optimizer. Therefore, this paper chooses the tf.train. AdamOptimizer to optimize the network model.

## Evaluation standard

The feasibility of the experiment and the detection performance of pulmonary nodules needs to be judged by objective evaluation parameters. Accuracy (ACC), sensitivity (SN), specificity (SP), free-response receiver operating characteristic (FROC) and area under the curve (AUC) were used to evaluate the experimental results [37]. The related concepts are described as follows:

ACC shows the probability that all samples are correctly detected. SN represents sensitivity, as well as the positive rate, and it represents that all positive samples are correctly detected. SN represents the ratio that true patients were accurately identified as sick. If SN was higher, the image with nodules had a greater probability to be detected and the misdiagnosis rate would be lower. SP represents specificity, as well as false positive rate, and it represents all negative samples are correctly detected. SP represents the ratio that no sick persons were accurately identified with no disease. If SP was higher, the image without nodules had a greater the probability to be detected and the rate of missed diagnosis would be lower.

The FROC curve can visually evaluate the test results, where the abscissa represents SP and the ordinate represents SN. The closer to the left upper corner it is, the better the detection performance. The FROC curve was not affected by the change in the distribution of positive samples and negative samples and can evaluate abnormal behavior, which is better than the traditional ROC curve. Another evaluation index is the AUC, where AUC is the area under the FROC curve, and AUC ranges from 0 to 1. The larger the value is, the better the detection performance.

**Table 6. The performance comparison of different activation functions.**

| activation functions | accuracy rate(%) | loss value |
|---|---|---|
| Sigmoid | 69.677 | 0.633933 |
| Tanh | 72.679 | 0.617793 |
| ReLu | 92.678 | 0.456618 |
| Elu | 89.478 | 0.465732 |
| ReLu6 | 63.926 | 0.683799 |
| Softplus | 53.997 | 0.506501 |

**Table 7. The performance comparison of different optimizers.**

| different optimizers | accuracy rate(%) | loss value |
|---|---|---|
| tf. train. Adadelta Optimizer | 66.942 | 0.675365 |
| tf.train.GradientDescentOptimizer | 69.813 | 0.696632 |
| tf. train.AdagradOptimizer | 78.091 | 0.621320 |
| tf. train. AdamOptimizer | 86.677 | 0.456618 |

## Experimental results

The proposed scheme adopts the five-fold cross-validation method to predict the dataset. Examples of successfully detected pulmonary nodules of the 3D cube transverse section and different sizes are presented in Fig 12, where d is the diameter of the predicted nodules, and p is the accuracy of detection. It can be seen that this scheme not only has a good performance in the detection of middle- and large-sized nodules but also has an accuracy rate of over 70% for the detection of small-sized nodules, The experimental results show that the multiscale characteristic structure of the 3D U-Net network model proposed can effectively detect pulmonary nodules of different sizes. Table 8 shows the detection accuracy of pulmonary nodules of different sizes under different false positive rates. The experimental results show that the detection accuracy was above 70% when the false positive rate reaches 4, and the detection accuracy tends to be stable. This scheme shows that it does not sacrifice the false positive rate to improve the detection accuracy and has strong availability.

## Comparison of experimental results

To further evaluate the performance of the proposed scheme, our experimental results were compared with existing 2D convolutional neural network [38] and 3D convolutional neural network [39]. The comparative experimental results are shown in Table 9. The results show that the accuracy, true positive rate and false positive rate of the 3D U-Net network model trained by transfer learning are better than those of 2D and 3D network models. The misdiagnosis rate and missed diagnosis rate are also effectively reduced, and the AUC area value was also the largest. The FROC curves of three different network models are shown in Fig 13. With the increase of training times and false positives, the detection performance of the proposed network model is better than that of the other two models. After the false positives reach 4, the detection rate tends to be stable. The experimental results show that the proposed network structure has a better ability to detect pulmonary nodules.

At the same time, in order to verify the effect of the performance in the detection of pulmonary nodules due to fine-tuning different layers' network parameters, experiments were designed to fine-tune network parameters of different layers. The experimental results are shown in Table 10. The results show that eight groups of different detection results can be obtained by adjusting network parameters of different layers. The results show that the detection accuracy of updating the parameters of the Conv5 layer was the worst, and the detection performance of updating Up6-Up8 layer parameters was better than other experimental groups. It can be concluded that the detection performance of updating the higher-level parameters was better than updating the lower-level parameters. The low-level information will detect more specific feature information such as lines and borders, etc., and the higher the level is, the more the abstract information can be detected. Because pulmonary nodules have complex features and rich high-level semantics, they are more helpful for the detection results that choose to update the high-level information.

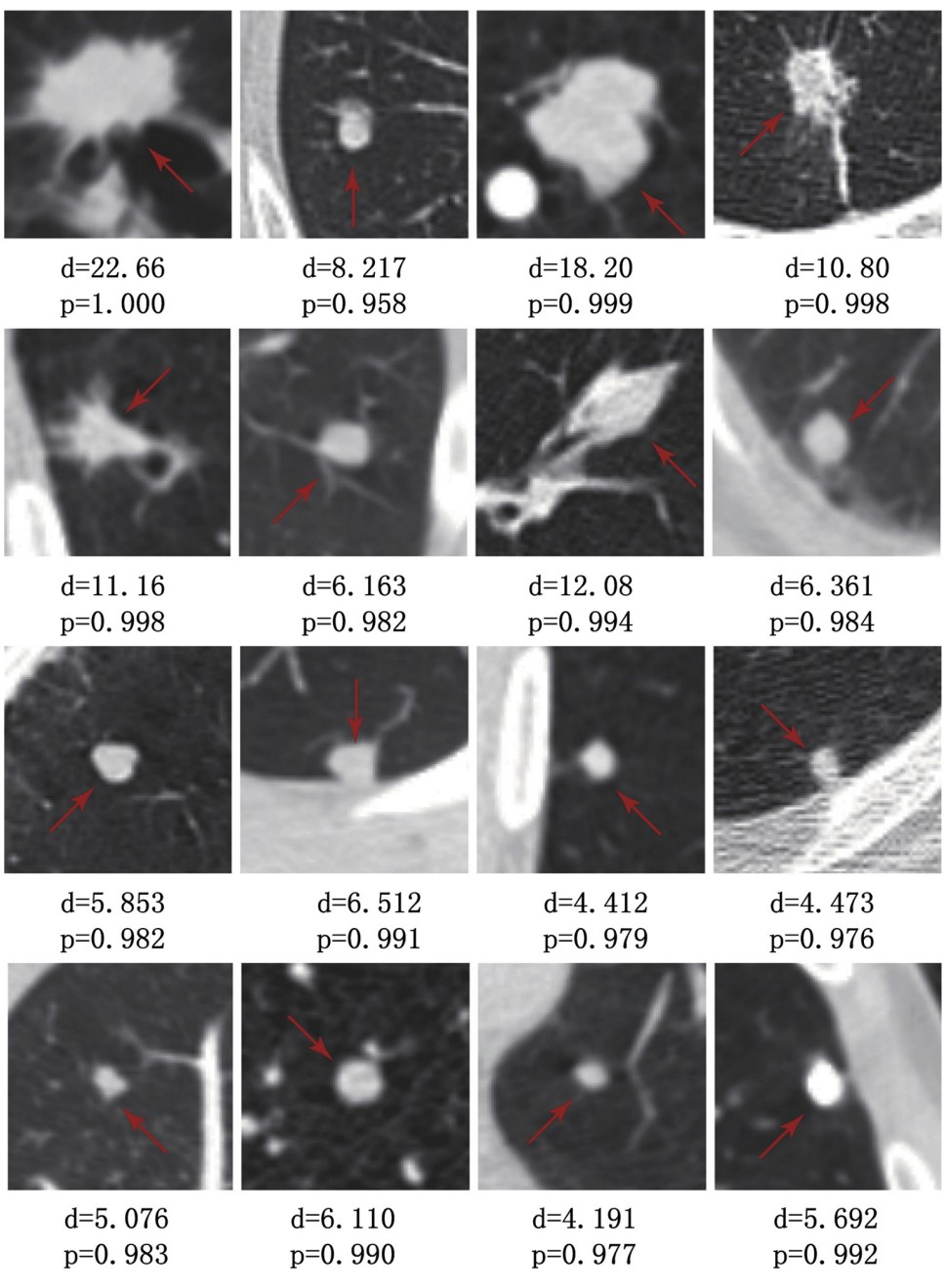

**Fig 12.**

To further evaluate the feasibility of the proposed scheme, this scheme was compared with other existing detection methods [40–49]. The evaluation indexes include SN, SP, false positive (unit FP/scan), and AUC. The experimental comparison results are shown in Table 11. The experimental results showed that the detection performance of the deep learning method was better than that of the traditional method. Khosravan's scheme [49] added dense connection blocks to avoid the loss of low-level information, and the overall detection performance was the best. Zhu's scheme [45] added the dual-path block and U-Net decoder in the 3D convolutional neural network, which could obtain more detailed nodule features, improved detection

**Table 8. Accuracy pulmonary nodules of different scales in different false positive rates.**

| Pulmonary nodule size(mm) | 0.125 | 0.25 | 0.5 | 1 | 2 | 4 | 8 |
|---|---|---|---|---|---|---|---|
| d = 4.125 | 0.4173 | 0.4258 | 0.5741 | 0.6058 | 0.6877 | 0.7065 | 0.7066 |
| d = 4.831 | 0.4291 | 0.4401 | 0.5833 | 0.6136 | 0.6971 | 0.7579 | 0.7579 |
| d = 5.760 | 0.4534 | 0.4593 | 0.5972 | 0.6642 | 0.7210 | 0.7953 | 0.7952 |
| d = 7.989 | 0.5020 | 0.4999 | 0.6341 | 0.6980 | 0.7432 | 0.8165 | 0.8100 |
| d = 9.125 | 0.5222 | 0.5387 | 0.6735 | 0.7096 | 0.7811 | 0.8532 | 0.8671 |
| d = 13.630 | 0.5651 | 0.6029 | 0.7151 | 0.7390 | 0.8290 | 0.9062 | 0.9060 |
| d = 14.192 | 0.5963 | 0.6672 | 0.7363 | 0.7431 | 0.8561 | 0.9115 | 0.9110 |
| d = 18.773 | 0.6290 | 0.6982 | 0.7490 | 0.7671 | 0.8894 | 0.9253 | 0.9253 |
| d = 24.241 | 0.6324 | 0.7140 | 0.7624 | 0.7972 | 0.9032 | 0.9314 | 0.9395 |

sensitivity, and the overall detection effect was the second best. Dou's scheme [42] proposed a 3D full convolutional network model of mixture loss to detect pulmonary nodules and obtained high specificity. Hao's scheme [46] and Broyelle's scheme [47] introduced the residual network structure in deep learning and obtained good detection results. On the basis of large data, this scheme obtained good performance values, and the overall performance was better than traditional methods. Multiscale feature structure and transfer learning training were used, and the evaluation indexes were higher than the existing deep learning methods. A large number of experiments showed that this scheme could automatically and accurately detect pulmonary nodules on the basis of high sensitivity and a low false positive rate.

## Conclusions

According to the characteristics of pulmonary nodules, this paper presented an automatic method for detecting pulmonary nodules. The experimental results showed that the algorithm had high accuracy in detecting pulmonary nodules of different sizes. For the small number of labeled images in the database of pulmonary nodules, the idea of transfer learning was introduced into the deep network model, and the features of the image were extracted by fine-tuning the structure of the network model. The method solved not only the problem that a small dataset was difficult to train a deep model but also the overfitting problem. The experimental results showed that layer by layer transfer training method improved the accuracy of image detection under the condition of small samples. It also saves the time that doctors use to manually mark large samples, and it can help doctors diagnose lung cancer more quickly and accurately. Data enhancement arithmetic was introduced, the stochastic gradient descent algorithm automatically updated the learning rate, and a series of simulation experiments were carried out on images of pulmonary nodules by changing the number of convolution layers, activation functions, optimization modes and pooling methods in the model. The ability of feature extraction and overfitting resistance of the network structure was improved. Finally, the proposed method was compared with the existing methods, and the feasibility and advantages of the proposed scheme were verified by experiments.

**Table 9. Results of different network models on dataset.**

| different network models | ACC | SN | SP | AUC |
|---|---|---|---|---|
| 2D convolution neural network [38] | 0.761 | 0.794 | 0.814 | 0.828 |
| 3D convolution neural network [39] | 0.874 | 0.868 | 0.882 | 0.925 |
| 3D U-Net convolutional neural network based on multi-scale feature of transfer learning | 0.968 | 0.924 | 0.946 | 0.941 |

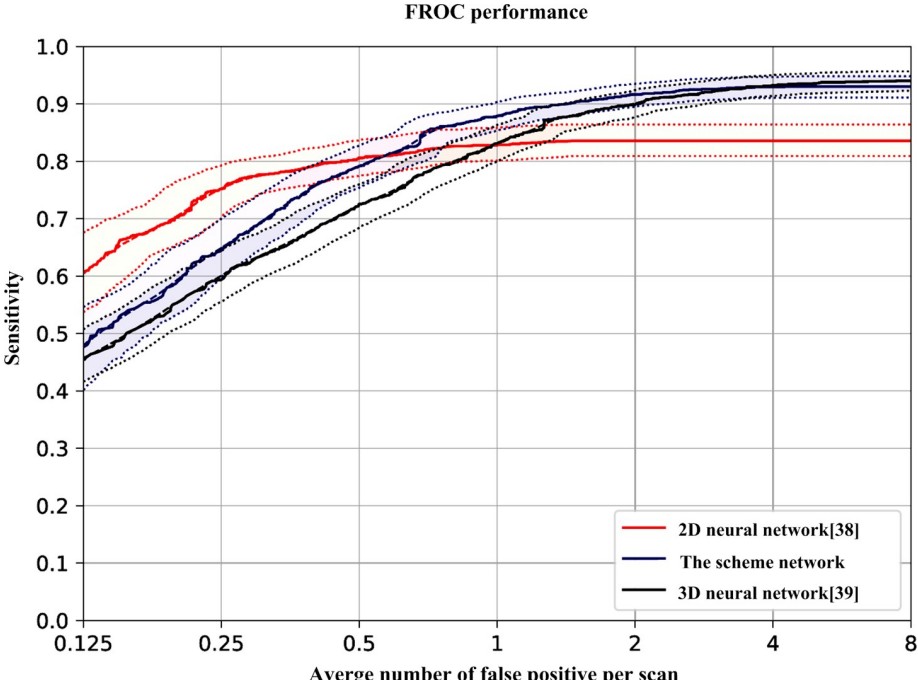

**Fig 13.**

## Limitation and expectation

In the detection of pulmonary nodules, this scheme could detect larger diameters of independent solid pulmonary nodules, smaller pulmonary nodules, or those adhering to blood vessels and tissue pulmonary nodules. However, some pure ground glass nodules (PGGNs) were missed, as shown in Fig 14. Unlike solid nodules, PGGNs do not have any solid components, which presented a small contrast between the brightness and the surrounding lung environment in the lung CT image, so it was difficult to extract image features of PGGNs. At the same time, PGGNs were a minority in LUNA16 and TIANCHI17 datasets and had not been fully trained. Therefore, PGGNs were missed. In the next step, PGGN data will be collected for training to enhance the detection of these pulmonary nodules.

Based on the above research results, further exploration is needed in the following aspects:

(1) The data collected comes from an open dataset. Because of the different equipment, the effect and age of the collected images are different, so few experimental images meet the

**Table 10. Experimental results of fine-tuned network parameters of different layers.**

| structural layers | SP | SN | AUC | ACC |
|---|---|---|---|---|
| convl-Up12 | 0.914 | 0.794 | 0.928 | 0.899 |
| coZnv2-Up10 | 0.922 | 0.798 | 0.925 | 0.896 |
| conv3-Up8 | 0.927 | 0.801 | 0.921 | 0.891 |
| conv4-Up6 | 0.946 | 0.824 | 0.917 | 0.887 |
| conv5 | 0.949 | 0.828 | 0.911 | 0.883 |
| Up6-Up8 | 0.968 | 0.877 | 0.945 | 0.929 |
| Up9-Up11 | 0.963 | 0.862 | 0.941 | 0.915 |
| Up12 | 0.957 | 0.866 | 0.937 | 0.903 |

**Table 11. Comparison of performance of the proposed pulmonary nodule detection scheme with some existing schemes.**

| Author | Dataset | SN(%) | SP(%) | False positive | AUC | Algorithm type |
|---|---|---|---|---|---|---|
| Wang et al. [40] | Private scans:103;nodules:127 | 82.6 | 73.4 | 4.0 FP/scan | 0.83 | Hybrid method |
| Setio et al. [41] | Private scans:158;nodules:183 | 81.1 | 79.7 | 5.45 FP/scan | 0.82 | Statistical pattern recognition |
| Dou et al. [42] | Public LIDC-IDRI;scans:648 | 86.5 | 93.9 | 4.3 FP/scan | 0.90 | Deep learning |
| Akram et al. [43] | Public LIDC-IDRI;scans:183 | 85.1 | 83.5 | 3.8 FP/scan | 0.83 | Statistical pattern recognition |
| Gong et al. [44] | Public LUNA16;scans:888 | 79.3 | 84.6 | 2.8 FP/scan | 0.90 | Morphology structure |
| Zhu et al. [45] | Public LIDC-IDRI;scans:391 | 88.6 | 90.9 | 4.7 FP/scan | 0.91 | Deep learning |
| Hao et al. [46] | Public TIANCHI17;scans:800 | 87.1 | 90.3 | 2.9 FP/scan | 0.92 | Deep learning |
| Broyelle [47] | Public LIDC-IDRI;scans:422 | 86.2 | 89.9 | 5.7 FP/scan | 0.91 | Deep learning |
| Gupta et al. [48] | Public LIDC-IDRI;scans:800 | 83.5 | 89.2 | 6.6 FP/scan | 0.84 | Hybrid method |
| Khosravan et al. [49] | Public LIDC-IDRI;scans:662 | 90.7 | 91.5 | 2.2 FP/scan | 0.93 | Deep learning |
| Proposed method | Public TIANCHI17 and LUNA16;scans:1688 | 92.4 | 94.6 | 4.0 FP/scan | 0.94 | Transfer and deep learning |

requirements. Therefore, collecting image sets of current equipment is the key task of future research. In the future, cross-analysis can be carried out with clinical medical image data of different types to enhance the generalization ability of the algorithm.

(2) In this paper, a method of combining transfer learning and a convolutional neural network was proposed to detect pulmonary nodules, although it was a new method with certain prospective. With the rapid development of transfer learning and the diversification of deep learning models, increasingly more new network models are applied to the

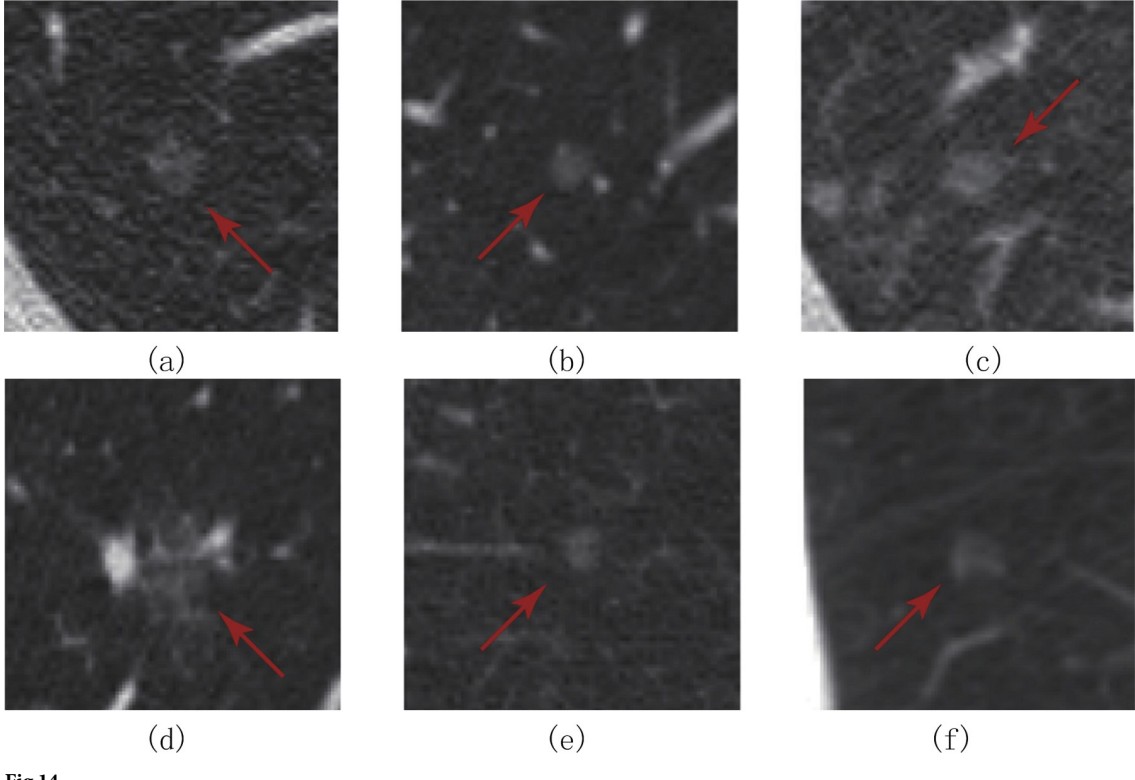

(a) (b) (c)

(d) (e) (f)

**Fig 14.**

detection of pulmonary nodules, which will be a trending topic in the field of medical image processing in the future.

(3) The proposed 3D U-Net model was only fine-tuned and trained to detect pulmonary nodules in the paper. The transfer learning idea is not applied to other existing network models for experimental comparison. Therefore, it is necessary to find and verify the most suitable depth model for detection of pulmonary nodules through both experiments and theory.

(4) Pulmonary nodules are the early manifestation of lung cancer. Detection of pulmonary nodules is only the first step in the early detection of lung cancer. It is necessary to further judge the benign and malignant features of nodules. More detailed studies of nodules are needed. For example, whether the surface is smooth, whether the interior is calcified, and whether it is hollow or solid need to be analyzed in detail, and nodules need to be accurately classified [50]. Therefore, further feature analysis and classification of the detected nodules are very important development directions in the future.

## Supporting information

**S1 Dataset.**
(DOCX)

## Author Contributions

**Data curation:** Siyuan Tang.

**Methodology:** Siyuan Tang.

**Validation:** Jinniu Bai.

**Writing – original draft:** Siyuan Tang.

**Writing – review & editing:** Min Yang.

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
