## [Decision Letter · Decision Letter 0]

27 Apr 2020

PONE-D-19-32789

Detection of Pulmonary Nodules based on a Multiscale Feature 3D U-net Convolutional Neural Network of Transfer Learning

PLOS ONE

Dear Siyuan,

Thank you for submitting your manuscript to PLOS ONE. After careful consideration, we feel that it has merit but does not fully meet PLOS ONE’s publication criteria as it currently stands. Therefore, we invite you to submit a revised version of the manuscript that addresses the points raised during the review process.

We would appreciate receiving your revised manuscript by May 16 2020 11:59PM. To enhance the reproducibility of your results, we recommend that if applicable you deposit your laboratory protocols in protocols.io, where a protocol can be assigned its own identifier (DOI) such that it can be cited independently in the future. For instructions see: http://journals.plos.org/plosone/s/submission-guidelines#loc-laboratory-protocols

We look forward to receiving your revised manuscript.

Kind regards,

Wajid Mumtaz

Academic Editor

PLOS ONE

Journal Requirements:

2. Please amend the manuscript submission data (via Edit Submission) to include authors Min Yang Jinniu Bai and Zhifang Pan

Reviewers' comments:

Reviewer's Responses to Questions

**Comments to the Author**

1. Is the manuscript technically sound, and do the data support the conclusions?

Reviewer #1: Yes

2. Has the statistical analysis been performed appropriately and rigorously? 

Reviewer #1: Yes

3. Have the authors made all data underlying the findings in their manuscript fully available?

Reviewer #1: Yes

4. Is the manuscript presented in an intelligible fashion and written in standard English?

Reviewer #1: Yes

5. Review Comments to the Author

Reviewer #1: General Comments

General:

In this paper, the authors reported a new computer-aided detection scheme is proposed, the 3D U-Net convolutional neural network to automatically detect pulmonary nodules from the thoracic region containing background and noise. In this work, the method solved not only the problem that a small dataset was difficult to train a deep model but also the overfitting problem. Moreover, layer by layer transfer training method improved the accuracy of image detection under the condition of small samples. The topic of the manuscript is interested and useful for audience. The reviewer only has some suggestions as follows:

Specific comments

1. Why only choose to rotate the image at 90°,180°,270°degrees in data enhancement?

2. Can you get better results if you use shifting, shearing other methods in the data enhancement part?

3. In Fig. 7, how to determine the parameters of 3D U-Net model have been trained properly, and can be transfer to another model?

4. Why choose 3D U-NET instead of other 3D CNN models?

5. In page 14, the explanation of cascading layer could better repair the details of the target should be more clarified.

6. Please rearrange the order of the figure (ex. Fig 9(c) 9(d)?).

6. PLOS authors have the option to publish the peer review history of their article (what does this mean?). If published, this will include your full peer review and any attached files.

Reviewer #1: No

---

## [Author Response · Author response to Decision Letter 0]

28 May 2020

1.Why only choose to rotate the image at 90°,180°,270°degrees in data enhancement?

The rotation Angle of the specification is easy to achieve algorithmically, and it also achieves the purpose of obtaining more images from different angles.

2.Can you get better results if you use shifting, shearing other methods in the data enhancement part?

A shift can increase the amount of data, Since most pulmonary nodules are small in size, incomplete pulmonary nodules are easily obtained by shearing.

3.In Fig. 7, how to determine the parameters of 3D U-Net model have been trained properly, and can be transfer to another model?

The stochastic gradient descent optimization method (SGD) is used to fine-tune the weight, the network model can obtain more features of pulmonary nodules. the adjusted network model can be transfer to the new data set to detect pulmonary nodules.

4.Why choose 3D U-NET instead of other 3D CNN models?

The improved multi-scale 3D U-NET network model can solve the problem of missing the low-level characteristics of pulmonary nodules in the process of network transmission, High-level features are acquired by convolution and pooling operations,high speed flow of feature information between input and output layers is achieved by using dense network and multi-scale features are generated combining with expanded convolution to improve the utilization of low-level features of pulmonary nodules,The detection accuracy of small nodules are also improved, so this paper used 3D U-NET model to detect pulmonary nodules.

5.In page 14, the explanation of cascading layer could better repair the details of the target should be more clarified.

The cascade layer merges the output layer of the first layer with the input layer of the second, It's easy to lose the boundary pixel information in the convolution operation, The cascade layer can obtain more feature information of the target area through the combination of the high-level feature and the low-level feature, which can achieve the purpose of better repairing the target details.

6.Please rearrange the order of the figure (ex. Fig 9(c) 9(d)?).

The figure has been rearranged.

---

## [Decision Letter · Decision Letter 1]

22 Jun 2020

Detection of Pulmonary Nodules based on a Multiscale Feature 3D U-Net Convolutional Neural Network of Transfer Learning

PONE-D-19-32789R1

Dear Dr. Jinniui Bai,

We’re pleased to inform you that your manuscript has been judged scientifically suitable for publication and will be formally accepted for publication once it meets all outstanding technical requirements.

Kind regards,

Wajid Mumtaz

Academic Editor

PLOS ONE

Additional Editor Comments (optional):

Reviewers' comments:

Reviewer's Responses to Questions

**Comments to the Author**

1. If the authors have adequately addressed your comments raised in a previous round of review and you feel that this manuscript is now acceptable for publication, you may indicate that here to bypass the “Comments to the Author” section, enter your conflict of interest statement in the “Confidential to Editor” section, and submit your "Accept" recommendation.

Reviewer #1: All comments have been addressed

2. Is the manuscript technically sound, and do the data support the conclusions?

Reviewer #1: Yes

3. Has the statistical analysis been performed appropriately and rigorously? 

Reviewer #1: Yes

4. Have the authors made all data underlying the findings in their manuscript fully available?

Reviewer #1: Yes

5. Is the manuscript presented in an intelligible fashion and written in standard English?

Reviewer #1: Yes

6. Review Comments to the Author

Reviewer #1: In the paper entitled "Detection of Pulmonary Nodules based on a Multiscale Feature 3D U-Net Convolutional Neural Network of Transfer Learning", the authors answered and added some contents according to reviewer’s suggestions on a point-by-point basis. In view of the scope, novelty, and quality of this work, I would recommend this paper to be published in Plos One.

7. PLOS authors have the option to publish the peer review history of their article (what does this mean?). If published, this will include your full peer review and any attached files.

Reviewer #1: No

---

## [Editor Report · Acceptance letter]

1 Jul 2020

PONE-D-19-32789R1 

Detection of Pulmonary Nodules based on a Multiscale Feature 3D U-Net Convolutional Neural Network of Transfer Learning 

Dear Dr. Bai:

I'm pleased to inform you that your manuscript has been deemed suitable for publication in PLOS ONE. Congratulations! Your manuscript is now with our production department. 

Kind regards, 

on behalf of

Dr. Wajid Mumtaz 

Academic Editor

PLOS ONE